# Teenagers' Awareness about Local Vertebrates and Their Functions: Strengthening Community Environmental Education in a Mexican Shade-Coffee Region to Foster Animal Conservation

**Ellen Andresen** [1] , **Paulina López-del-Toro** [1], **Montserrat Franquesa-Soler** [1,*] ,
**Francisco Mora** [1] **and Laura Barraza** [2]

[1] Instituto de Investigaciones en Ecosistemas y Sustentabilidad, Universidad Nacional Autónoma de México (IIES–UNAM), Morelia 8701, Mexico; andresen@cieco.unam.mx (E.A.); lopez.paulina@gmail.com (P.L.-d.-T.); fmora@cieco.unam.mx (F.M.)

[2] Education for Sustainability, Servicios Ambientales, Conservación Biológica y Educación A.C. (SACBÉ), Mexico City 04010, Mexico; laubarraza@gmail.com

*  Correspondence: franquesamontse@gmail.com

**Abstract:** Peoples' understanding and appreciation of wildlife are crucial for its conservation. Nevertheless, environmental education in many tropical countries is seldom incorporated into public-school curricula and wildlife topics are often underrepresented. In this research we aimed to (1) assess the effects of an environmental education intervention focused on improving students' awareness about wild vertebrates and their ecological functions and (2) to evaluate whether previous exposure to general environmental education could improve the effects of the intervention. We worked in four schools in a high-biodiversity shade-coffee-producing region in Mexico; two of the schools had received general environmental education as part of a Community Program, while the other two had not. In all schools we conducted a targeted intervention providing information about wild vertebrates and their ecological functions. Through questionnaires, we assessed students' awareness before and after the intervention. We found that students' awareness about wildlife was improved by our intervention, and that this effect was stronger in students that had attended the Community Program. Our results contribute to Sustainable Development Goals 11 and 15 by showing that targeted education interventions can help achieve specific conservation goals, and that previous community-based environmental education can condition peoples' awareness, improving the assimilation and/or understanding of new concepts.

**Keywords:** agroecosystem; biodiversity conservation; conservation education; indigenous community; local knowledge

## 1. Introduction

Since the United Nations launched the 17 Sustainable Development Goals (SDGs) in 2015, governments have made commitments to ensure an integral sustainable development of their countries, i.e., social, economic, and environmental sustainability [1]. To meet SDGs, education will need to play a crucial role [2]. This is recognized in the Education for Sustainable Development framework, which encompasses the pedagogic responses and the set of competencies needed to deal with global challenges in daily life [2]. Environmental sustainability in particular requires a change in human behavior, and a starting point is to use environmental education to improve peoples' understanding of environmental problems and the ways to ameliorate them. However, linking global and national goals

to local initiatives is not an easy task due to the complexities of each unique regional context. Thus, the success of environmental education depends on our ability to implement it effectively in culturally sensitive and context-specific ways [3], i.e., considering local agendas first in a bottom-up approach towards achieving environmental sustainability.

The implementation of environmental education is particularly urgent in tropical countries, which harbor the most biodiversity on the planet [4] but which also suffer serious socio-economic problems that greatly endanger this biodiversity [5]. To succeed in such complicated contexts, conservation and sustainable management efforts should consider environmental education programs that build upon culturally meaningful connections between local livelihoods and nature [6].

Animals constitute one of the most threatened components of tropical biodiversity [7] while at the same time being crucial providers of ecosystem functions and services [8]. Yet the decline in wildlife populations is rarely perceived by decision makers as a crucial driver of global change, but rather as merely a consequence of it [9]. Neglecting to consider the important roles that wildlife play in ecosystem function threatens the long-term success of widely implemented forest conservation initiatives such as REED+ (Reduced Emissions from Deforestation and Degradation REDD+ [10]). Given that hunting and illegal trade are two of the major threats to wildlife [11,12], people living in rural areas play a key role in determining the fate of animals inhabiting the countryside surrounding their communities. Their fate can range from negative (e.g., [13]) to positive (e.g., [14]), depending on the attitudes of local people. In turn, peoples' attitudes are related to their knowledge and perceptions of that wildlife [15].

Knowledge and perceptions acquired during childhood and adolescence are thought to affect lifelong behavior and attitudes towards the environment [16]. In particular, students at the secondary level of education are considered good candidates for developing ethical and ecological appreciation of the natural world [17,18]. This is a critical stage, since teenagers' beliefs are typically shaped and consolidated between the ages of 10 and 16 [19]. Additionally, for many individuals from tropical countries living in rural areas, this is the last time they engage in formal education [20]. Moreover, the voices and acts of young people can turn them into change agents and raise consciousness in the context of the current biodiversity crisis [21,22]. Regarding wildlife, several studies found that young people can have a strong affinity towards animals but that this positive connection can be lost over time unless reinforced in a meaningful way [6,23].

What can environmental education programs do to achieve such meaningful reinforcement? First, these programs should not simply use information and tools applied to different contexts. Instead, they ought to design context-specific contents aligned to students' local realities, using their existing knowledge and perceptions as a starting point [24]. Second, environmental education should be a part of all school curricula, implemented at all levels (i.e., from pre-school to the last year of secondary school) and in all subjects as a transdisciplinary axis. Studies show that repeated exposure to relevant information can have a "build-up effect" that increases and strengthens environmental awareness [25,26]. Unfortunately, most tropical countries do not incorporate environmental education into their public-school curricula. Thus, in many biodiversity-rich countries, conservation education almost entirely depends on local initiatives and/or projects implemented by non-governmental conservation organizations, the effects of which often remain undetermined.

The main goal of our study was to implement an education intervention targeted towards improving wildlife awareness. Our study is aligned with the United Nation's Sustainable Development Goal 11 (SDG 11), which is to foster sustainable and resilient human settlements, and SDG 15, which aims to halt biodiversity loss. We worked with secondary schools in a high-biodiversity shade-coffee-producing region in Mexico, where an environmental education Community Program had been implemented in some schools, but not others. This program addressed general environmental aspects related to the sustainable use of natural resources, with relatively little emphasis on wildlife. We designed and implemented an education intervention that provided content- and context-specific information about vertebrates (medium/large mammals, bats, birds, and snakes) and their ecological

functions in the local ecosystems (pollination, seed dispersal, biological control). With our study we aimed to answer two specific questions: (1) Do the targeted education intervention and/or the Community Program improve the awareness that students have about the local wildlife? and (2) Does previous exposure to general environmental education (i.e., the Community Program) strengthen the effect of the intervention? We expected that our intervention would improve students' awareness about animals and their ecological functions. Although the Community Program did not focus on animals, some topics related to wildlife were nonetheless discussed. Thus, we expected some improvement in awareness due to the Community Program itself. We also expected that information previously acquired through the Community Program could facilitate the assimilation of new concepts, and thus boost the effect of our intervention.

## 2. Materials and Methods

### 2.1. Study Area, Community Program, and Participants

The study was conducted in 2006 and 2007 in the municipality of Cuetzalan, in the northeastern part of the Mexican state of Puebla (Figure 1; [27]).

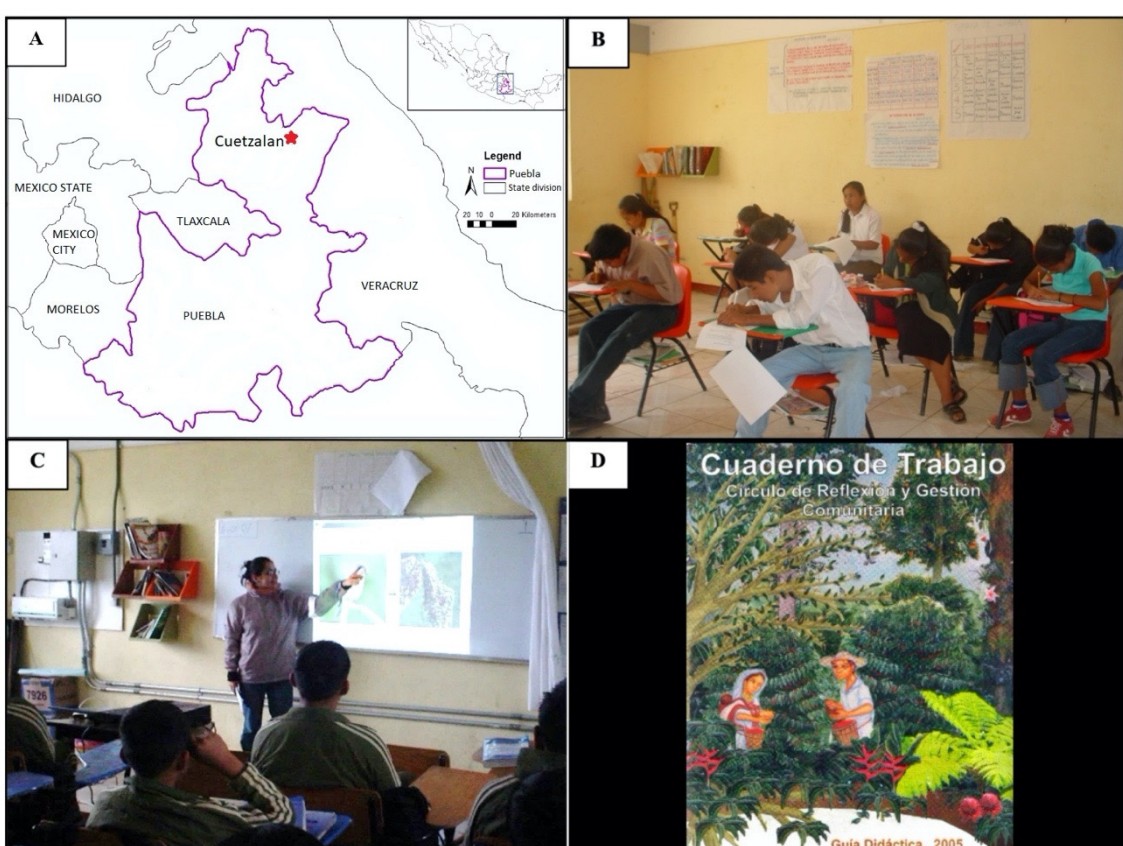

**Figure 1.** (**A**) A map showing the location of the study site, Cuetzalan municipality, in the Mexican state of Puebla. (**B**) Some of the participant students answering the questionnaires used during the pre- and post-intervention evaluations. (**C**) An oral presentation with slides was part of our intervention. (**D**) An especially designed workbook was used to guide the workshops carried out by the Community Program. Photos in (**B**,**C**): Paulina López-del-Toro.

The region has three types of natural vegetation: pine-oak forest, tropical rainforest, and cloud forest, although the latter has been almost completely replaced by agricultural crops (most importantly traditional shade coffee) and cattle pastures [28]. The municipality has a high diversity of vertebrates:

54 species of reptiles and amphibians, 115 species of birds, ~20 species of terrestrial mammals (excluding small rodents and marsupials), and more than 30 species of bats [28–30].

During the study period, Cuetzalan had a population of 45,781 [27] and the main economic activity was agriculture. The indigenous cooperative Tosepan Titataniske (meaning "together we will succeed" in the indigenous Náhuatl language) is an important local organization whose main activity has been the production of shade coffee since 1978, with an emphasis on organic shade coffee grown in highly diverse traditional systems [31]. When the study took place, the cooperative had ~5800 members belonging to 60 communities and had launched several programs with the main goal of improving quality of life [32]. Today, the organization has grown to form a regional network of cooperatives and includes 34,000 families (mostly indigenous groups of Nahuas and Tutunakus) from 26 different municipalities [33].

One of the community programs that the cooperative Tosepan Titataniske started in 2004 with the assistance and funding of an external start-up project focused on environmental education [31,34]. This program was implemented to emphasize the importance of natural resources through workshops held both with farmers and with students in local schools. A workbook was specially designed as a tool for carrying out the workshops ([35]; Figure 1). The workbook addressed various issues, but most of the emphasis was on sustainable practices for promoting environmental conservation in culturally meaningful ways [31]. The goal of the start-up project was that the cooperative would fully take over and continue financing and implementing the program; unfortunately, this did not occur, and the program only lasted for approximately six years (P. Moguel, pers. comm.).

In schools, the program consisted of one Community Program educator working with each class for two hours daily for approximately two weeks, covering all the contents of the workbook. Regarding biodiversity, the main focus was on plants and relatively less of the workbook's content was about animals. The few topics related to wildlife were: (a) biodiversity includes more than just plants, (b) animals that can be found on shade-coffee plantations, (c) roles that insects, mammals, birds, amphibians, and reptiles play on shade-coffee plantations, (d) actions that harm vs. help conserve biodiversity in shade-coffee plantations (the list of 29 actions included "hunting birds", "being respectful of animal reproductive cycles", and "eliminating all insects").

We worked with 120 students aged 12–15 from four secondary schools in villages located close (4–5 km) to the city of Cuetzalan. Most students (80%) had parents who were members of the Tosepan cooperative. Half of the students attended schools that had participated in the environmental education Community Program at some point during the previous two years (Leonardo Bravo school in Tuzamapan/Xiloxochico villages and Carlos Dickens school in Acaxiloco village); the other half attended schools that not participated in the Community Program (Héroes del 5 de Mayo school in Pepexta village and José Trinidad Salgado León school in Xalpantzingo village). We obtained oral permission from the school principals, teachers, and the Tosepan cooperative. We worked in collaboration with the Community Program educators and our study was introduced as a follow-up to that program.

*2.2. The Intervention*

To provide specific information on local vertebrates and their ecological functions, we designed an environmental education intervention that was implemented once in each school in January 2007 by one of us (P.L.-d.-T.) with the assistance of one of the Community Program educators (Figure 2). Local teachers were present during the activities and were genuinely interested in the information we provided. The intervention included: (a) an oral presentation of approximately 45 min (with 60 slides as a visual aid) during which the students were encouraged to actively participate (Figure 1) and (b) distribution of a three-page leaflet [36]. A leaflet was given to each student and read out loud in the classroom after the presentation to reinforce the message and address any additional questions and/or comments. During the presentation and in the leaflet, the following topics were addressed: (1) ecological functions of fauna (pollination, seed dispersal, predation, and biological control);

(2) myths and superstitions about wildlife; (3) natural history and functions of several selected groups of vertebrates: birds, bats, snakes, and particular mammals present in the area (skunks, opossums, martens, armadillos, raccoons, foxes, water dogs, kinkajous, etc.); and (4) extinct and threatened local vertebrates, main threats to animals, and actions to help conserve them.

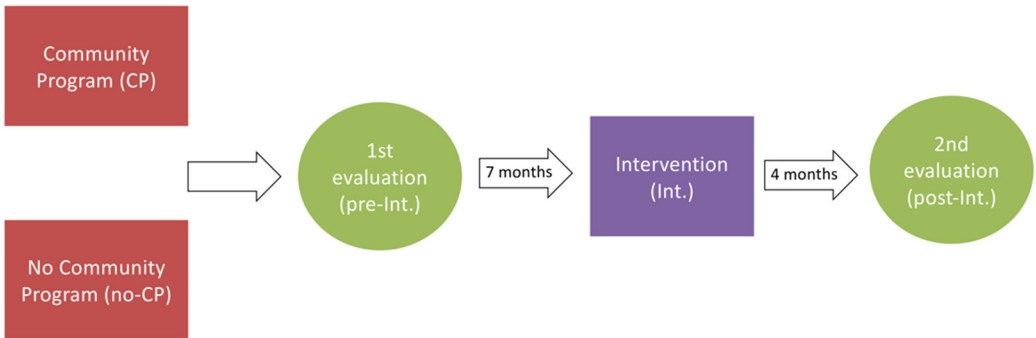

**Figure 2.** Sequence of the whole educational process. Two groups of students—those attending schools that participated in the general environmental-education Community Program (CP) and those attending schools that did not participate in the Community Program (no-CP)—were evaluated with the same questionnaire twice: the first evaluation took place seven months before our intervention (pre-Int.) and the second evaluation took place four months after our intervention (post-Int.). Our intervention provided content-specific information about wild vertebrates and their ecological functions, i.e., information that received little emphasis in the Community Program. The questionnaire used in both evaluations focused on the awareness (i.e., knowledge and perception) that students had on wild vertebrates and their ecological functions.

*2.3. Evaluation: Pre- and Post-Intervention Questionnaires*

To evaluate the effect of our intervention, we used a before-and-after study design. Prior to the intervention, in June 2006, we conducted a first evaluation through a questionnaire consisting of 19 questions (Table S1), which was applied to all students (Figures 1 and 2). Educators remained in the classroom but did not discuss answers with students. This first questionnaire also allowed us to evaluate the effect of the Community Program and to determine the baseline knowledge of the students. The questions assessed students' knowledge, or a combination of knowledge and perceptions, on wild vertebrates and their ecological functions (Table S1). All questions had three possible answer options, except for one that had four and one that had six. Students could only mark one answer, except for the question with six answer options, where they could mark all they thought applied. Answer options were assigned score values along an ordinal scale, so responses associated with higher awareness about wildlife were assigned higher score values. In four of the questions, two of the possible answers were assigned the same score value because they were considered to indicate a similar level of awareness. In the question with six answer options, the three correct answers were assigned a score of +1, while the three incorrect answers were assigned a score of −1; the final score for any given student was obtained by adding the scores of the answer options he/she marked (Table S1).

To evaluate the impact of our intervention and of the interaction between the Community Program and the intervention, we conducted a second evaluation by applying the same questionnaire again, to the same students, in May 2007, four months after the intervention (Figures 1 and 2). We acknowledge that we did not have an absolute control group, i.e., a group of students who did not receive the intervention, and that this somewhat limits our inference ability. However, during the time period between both questionnaires, the schoolteachers remained the same and no other educational or conservational campaigns took place. So, we believe that the before-after design we used, which considers each individual as its own control, allows us to associate changes in students' responses to our intervention.

### 2.4. Data Analyses

Of the 120 students who answered questionnaires before the intervention, we were able to apply the post-intervention questionnaire to 99 (82.5%; 58 of the 60 who had participated in the Community Program, and 41 of the 60 who had not); we only used data obtained from these students in our analyses. In all analyses we also included the gender of each student (male, female) as a covariable, since previous studies show that gender can affect the knowledge and attitudes of individuals towards wildlife (e.g., [37,38]).

All the analyses were run using the R program version 3.6.1 [39]. To assess the overall effects of the Community Program and of the intervention on students' awareness about wild vertebrates, we performed a distance-based redundancy analysis (db-RDA). As is the case with other asymmetric canonical ordination methods, the aim of a db-RDA is to model a multivariate response dataset (students' answers to the 19 questions) that is hypothesized to relate to a predictor dataset, and to formally test statistical hypotheses about the significance of those relationships [40]. Redundancy analysis combines multiple regression with classical principal component analysis (PCA) to perform such modelling, and the significance of such tests is assessed by means of permutations. Furthermore, db-RDA allows for the inclusion of different kinds of variables in the response matrix, including those measured on an ordinal scale, by previously transforming the response dataset to a distance matrix using any distance measure [41].

To perform the db-RDA on our data, we first transformed the matrix of students' answers to a distance matrix (D) using a symmetric Gower distance with the *gowdis* function in the FD package [42]. We then fitted the db-RDA with $D$ as the response and Community Program (CP), intervention (Int.), and their interaction (CP × Int.) as predictors; we also included gender as a covariable. The procedure was implemented using the *capscale* function, and the statistical significance of the predictors and covariable was tested using the pseudo-F statistic as calculated by the *anova.cca* function, both in the vegan package [43]. Significance tests were based on 1000 permutations that took into account the aggregated nature of the dataset: Responses from the same individual and individuals from the same school were not dissociated during the randomization process.

Then, to further explore the effects of the Community Program and the intervention on the answers given to each specific question, we fitted linear mixed models for ordinal variables using the *clmm* function in the ordinal package [44]. These models assess how the probability of different responses changes as a function of a series of predictors. As in the db-RDA, we included Community Program, intervention, and their interaction as predictors and controlled for gender differences. Individual and school were included as random factors, allowing for random variation in the intercept of the model in order to take into account the possible non-independence of the data from students from the same school or data taken from the same individual at different times. The significance of the predictors and the covariable were tested using the Wald test, whose statistic follows a Z distribution.

## 3. Results

We found that the students' awareness about wildlife was affected by the intervention (pseudo-$F_{1,193}$ = 2.31, $p < 0.001$) and by the interaction between the Community Program and the intervention (pseudo-$F_{1,193}$ = 1.13, $p = 0.009$; Figure 3). Not only did the awareness of students improve after our intervention, but this change was also more pronounced in the group that had participated in the Community Program (Figure 3). In other words, our intervention was, on average, more successful in improving the awareness of a student towards wildlife if that student had had previous exposure to environmental education through the Community Program. Students' responses were not affected by the main effect of the Community Program (pseudo-$F_{1,193}$ = 1.45, $p = 0.330$). The covariable—gender—did not affect responses (pseudo-$F_{1,193}$ = 0.99, $p = 0.491$).

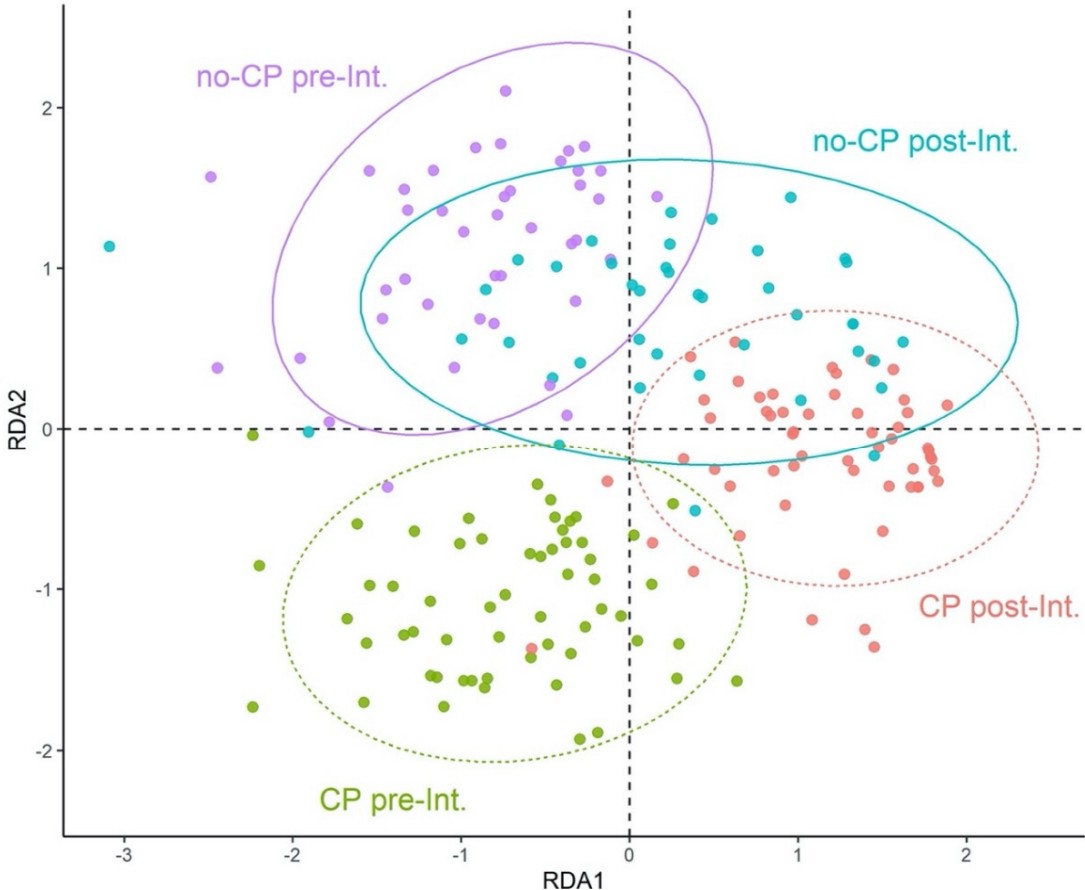

**Figure 3.** Results of the distance-based redundancy analysis (db-RDA). Colored dots represent students, each one characterized by his/her responses to the 19 questions from the evaluation questionnaire (Table S1). Each student is represented twice: once based on the responses given before our intervention (pre-Int., left part of the ordination graph) and once based on the responses given after our intervention (post-Int., right part of the ordination graph). Students attending schools that participated in the Community Program (CP) are in the lower part of the ordination graph, while those that attended schools that did not participate (no-CP) are in the upper part.

When questions were analyzed individually, we observed large variation among the patterns of students' responses (Figure 4). For many of the questions, answers that were associated with high awareness were the most common. For example, almost all students, both before and after the intervention, responded that wild animals that enter people's homes should not be killed, that caring for plants and animals is important for protecting the environment, and that snakes eat rodents (Figure 4D,E,I, respectively). On the other hand, questions for which the level of awareness tended to be lower were those related to the interdependence of plants and animals (Figure 4A,B), the one asking whether all snakes are venomous (Figure 4G), all questions about bats (Figure 4K–M,O), and two of the questions related to actions that threaten wild vertebrates (Figure 4C,R).

Students' responses to 12 of the 19 questions were significantly affected by one or more of the predictors of interest (Figure 4, Table S2). Though not statistically significant, responses to the remaining seven questions showed the expected tendencies (i.e., improved awareness in CP vs. no-CP students and/or in post-Int. vs. pre-Int. questionnaires). We found that only one question showed an effect from the covariable: Boys from both groups had better knowledge than girls when asked whether bats control insect pests (Table S2).

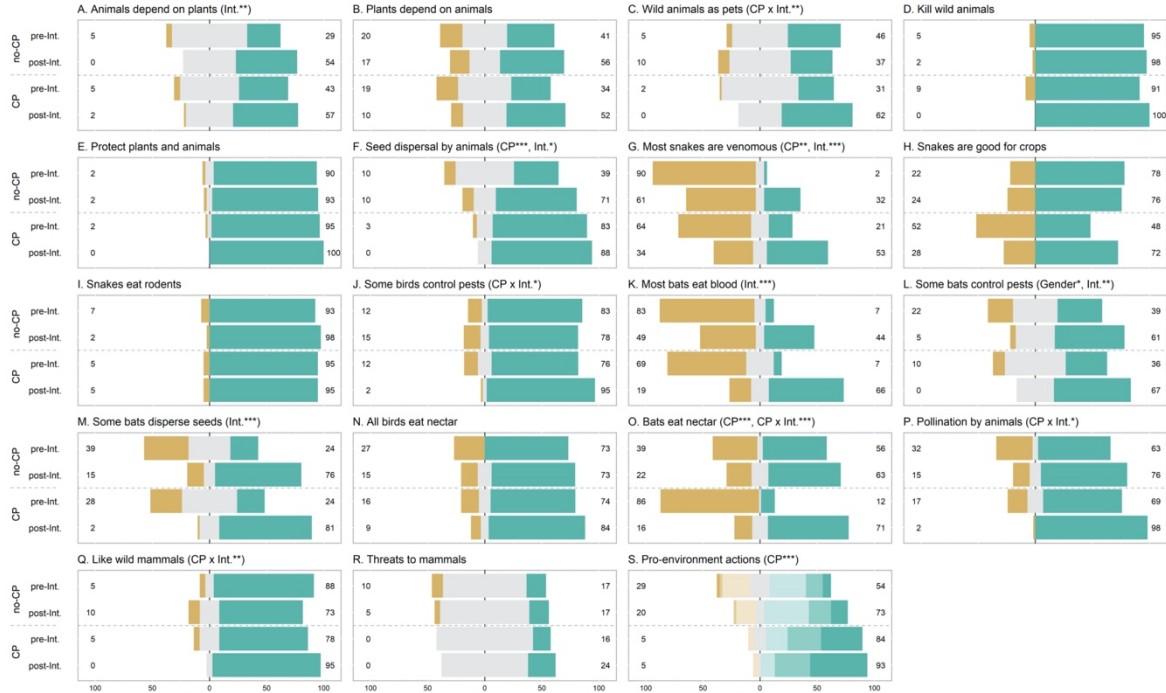

**Figure 4.** Responses to each of the 19 questions (A-S) in the evaluation questionnaire. Each bar represents the 100% of students answering a question. The four bars in each panel represent two different groups of students: those who had participated in the Community Program (CP) and those who had not (no-CP), and two different moments when each student answered the questionnaire: before the intervention (pre-Int.) and after the intervention (post-Int). The segments of different colors in each bar represent the percentage of students choosing answers with different scores; answers with scores indicating the highest level of awareness about wildlife are shown in green, answers indicating the lowest level are shown in orange-brown, and answers with intermediate scores are shown in grey. The numbers to the right and left of each bar show the percentage of students choosing the answer with the highest and lowest score value, respectively. Each question is identified by a short phrase in the upper part of its panel (**A**–**S**) that makes reference to the content of the question (full questions can be found in Table S1). Results of statistical analyses are shown in parenthesis on top of each panel, indicating which predictor(s) or covariable had a significant effect on the responses to that question: Community Program (CP), Intervention (Int.), interaction (CP × Int.), or gender. The full statistical output for the analyses of individual questions can be found in Table S2.

For four questions, students' responses were influenced by the Community Program. For three of these (importance of seed dispersal by animals, Figure 4F; whether all snakes are venomous, Figure 4G; and pro-environment actions, Figure 4S), students who had participated in the Community Program had higher awareness than students who had not participated in the Community Program. Interestingly, when asked whether bats eat nectar, students who did not attend the Community Program showed better knowledge, in particular before the intervention (Figure 4O).

Two of the questions that were positively influenced by the Community Program and four additional questions (six in total) were positively affected by the intervention to a similar degree in both groups of students (CP and no-CP). After the intervention, students showed higher awareness when asked whether animals depend on plants (Figure 4A), whether seed dispersal by frugivorous animals occurs and is important (Figure 4F), whether most snakes are venomous (Figure 4G), whether most bats feed on blood (Figure 4K), whether some bats control pests (Figure 4L), and whether some bats disperse seeds (Figure 4M). Additionally, the intervention improved the answers to five other questions, but differently depending on the group of students (i.e., significant CP × Int. interaction). In particular, when asked whether wild animals should be kept as pets (Figure 4C), whether birds control pests

(Figure 4J), and whether they liked wild animals (Figure 4Q), only the answers of students who had participated in the Community Program improved after the intervention. Also, when asked whether nectarivorous animals pollinate plants and whether this is good for the plants (Figure 4P), and when asked whether bats eat nectar (Figure 4O), both groups of students had improved answers after the intervention, but the change was much more pronounced in students who had participated in the Community Program.

## 4. Discussion

In this research, we found that our targeted intervention was effective in improving students' awareness of wildlife and their ecological functions, and that this effect was stronger when students had also received general environmental education through the Community Program. Our results underscore the fact that previous environmental education can condition peoples' awareness, leading to a positive synergic effect that improves the assimilation and/or understanding of new concepts. Our study also highlights the importance of designing custom-made educational programs that adapt to the local socio-ecosystem [45,46]. Finally, our study clearly emphasizes the important role that environmental education about animals and their ecological functions can play when implementing wildlife conservation projects in anthropogenic landscapes.

In the same way that aspects of the local context ought to be taken into account when planning environmental education programs, these aspects must also be considered when evaluating and interpreting the results of those programs. Thus, to better understand the results we obtained, we must recall that our study took place in rural communities that are part of a well-organized indigenous cooperative (Tosepan Titataniske), and where shade-coffee plantations are the basis of their livelihoods. These plantations are polycultures that are managed traditionally, providing habitat for a tremendous amount of biodiversity [47,48]. In fact, in the last two decades, traditional agroforestry systems such as the shade-coffee farms in our study site have emerged as the key element to conserving biodiversity under the land-sharing paradigm in anthropogenic tropical landscapes [49].

Most valued among the biodiversity components of the coffee farms in our study region are the ca. 300 wild and cultivated plant species that can be found in this agroecosystem, many of which are used in different ways, including food, medicine, fuel, insecticide, and crafts [47,48]. Since the projects implemented by the cooperative arose from local needs and interests [31], topics covered by the environmental education Community Program had a strong emphasis on useful plants and the sustainable management of the coffee farms [35]. Thus, although many groups of wild animals perform crucial ecological functions that are key to sustainable management of the coffee farms, the Community Program focused relatively less on these topics. This is the most likely reason why a positive effect from the Community Program was only detected in 3 of the 19 questions (Figure 4). One of these questions was about snakes being venomous, another was about general pro-environment actions that were stressed in the workbook used by the Community Program, and only one was related to the functional importance of animals (seed dispersal). By failing to explain and sufficiently stress the importance of wild animals for the functioning of ecosystems, both natural and anthropic, the Community Program missed an opportunity to improve the odds of wildlife conservation.

In contrast, our content-specific intervention clearly improved students' knowledge and perceptions in 11 out of the 19 questions about wild animals and their ecological functions. This learning process, in addition to filling a gap detected in the Community Program, was also culturally relevant, as students could relate fauna-derived services directly to the coffee farms, which play a central role in their realities [50]. This connection in turn increases the probability that the improved awareness might transcend and become pro-environmental behavior.

Environmental knowledge is usually considered a prerequisite for the adoption of pro-environmental behavior, but not a sufficient condition in itself [25,51]. Similarly, providing environmental information in itself may not be enough to improve people's knowledge. For example, despite the fact that our intervention provided concrete information about there being few venomous

snakes in the region and that hematophagous bats are very rare, and even though responses improved after the intervention, wrong answers were still very frequent (mainly for those who had not taken the Community Program; Figure 4G,K). These results are consistent with a related study in the same place, where the majority of shade-coffee farmers (61–73%) mentioned that most snakes were venomous and that most bats feed on blood [32], perceptions that are far from their reality. Our results and the previous findings underscore two crucial aspects of environmental education. First, that providing information may not suffice to improve knowledge, because people's perceptions and attitudes towards nature can be strongly affected by cultural context, traditions, and religious beliefs [37,52]. Environmental education needs to address emotional processes, disseminating information through an appeal to people's empathy, such that despite being unattractive, non-charismatic, or even feared, animals can be perceived as playing crucial roles in ecosystems and thus deemed worthy of conservation. And second, that certain knowledge, perceptions, beliefs, and/or attitudes can be transmitted intergenerationally, both from young individuals to adults (e.g., [53]) and vice versa (e.g., [54]). It is important that environmental education programs consider intergenerational transmission, as it can potentially boost the programs' outcomes or slow them down, depending on the information that is being transmitted.

Regarding the students' integral education, it is important to highlight that environmental education is generally addressed (when addressed at all) independently from a pedagogical perspective [55]. This means that new environmental concepts are not linked with significant previous experiences important to the students' context (e.g., community activities). The present study followed a constructivist learning model in which our intervention was integrated into a previous learning experience, the Community Program. Constructivist approaches help learners shape their own understanding by building upon their prior knowledge. In order to internalize information properly, it is important that new concepts make sense and are useful in the context of everyday life [55].

Finally, although it was not our goal to assess gender differences in student responses, it is worth mentioning that, with the exception of one question, gender had no effect on our results. This is interesting, because many studies have found a strong gender effect, with females generally showing more conservation-conducive responses (e.g., [38,56,57]). The cultural context of our study may have influenced this result. The long-term projects carried out by the Tosepan cooperative, in collaboration with different actors such as universities and non-governmental organizations (NGOs), could have positively affected the development of the students' values, promoting gender equality, sustainable practices, and peace education.

## 5. Conclusions

Our study highlights the positive effects that simple educational interventions can have when they are culturally relevant, particularly when they build upon the foundations of locally established programs. Longer-term programs, like the six-year Community Program, are more likely to cause real behavior change [58], and education based on the local indigenous heritage is a fundamental prerequisite for the development of culturally healthy communities [59]. The future of tropical forest biodiversity strongly depends on the effective and sustainable management of human-modified landscapes. Ultimately, people who live in these landscapes are the ones making decisions about whether to protect biodiversity or not. However, the long-term impacts of those decisions on their ecosystems and on their own nature-based economy are difficult to grasp when short-term subsistence is the daily pressing issue. To help achieve a deeper level of environmental awareness, the public education system of biodiversity-rich countries such as Mexico should include contextualized conservation education as an integral part of the curricula. But this will require a tremendous change that, unfortunately, is unlikely to occur soon in most tropical countries. Until this happens, the alliances between national and international NGOs, conservation scientists, and local authorities, are needed to design and implement environmental education programs that promote the conservation of biodiversity, including the often-overlooked animals, which are crucial for the functioning of tropical ecosystems.

**Supplementary Materials:** The following are available online at http://www.mdpi.com/2071-1050/12/20/8684/s1, Table S1: Questions presented in the evaluation questionnaire, their corresponding ID letter and phrase used in Figure 4, and their answer options with respective score values. We explained to students that the questions only addressed wild animals, and stressed that farm animals, pets, and invertebrates were excluded. Answer options were assigned score values on an ordinal scale, with lower scores representing a lower level of awareness and regarded as being less conducive to faunal conservation. Table S2: Parameter estimates of the ordinal logistic regression models adjusted to each question. Columns with the form "x|y" are the log-odds of "x" vs. "y" (where x and y are scores assigned to the answers; see Table S1), i.e., the likelihood of having a score ≤ x against having a score ≥ y. Positive values mean that scores ≤ x have higher probability, while negative values mean that scores ≥ y are more likely. Only question S had seven different alternative scores (−3 to 3), all others having three possible scores (1 to 3). Values in other columns provide a measure of the effect on these log-odds of changing from a basal level (Gender = Female, CP = No, Int. = pre-intervention) to an alternate level (Gender = Male, CP = Yes, Int. = post-intervention). CP = Community Program, Int. = Intervention.

**Author Contributions:** Conceptualization, P.L.-d.-T. and E.A.; methodology, P.L.-d.-T., E.A., M.F.-S. and F.M., software, F.M.; validation, E.A., M.F.-S. and L.B.; formal analysis, F.M.; investigation, P.L.-d.-T. and E.A.; resources, E.A.; data curation, P.L.-d.-T., E.A. and M.F.-S.; writing—original draft preparation, E.A. and M.F.-S.; writing—review and editing, E.A., M.F.-S. and L.B.; funding acquisition, E.A. All authors have read and agreed to the published version of the manuscript.

**Funding:** This project was funded by Mexico's Consejo Nacional de Ciencia y Tecnología through a research grant to E.A. (CONACyT 2005-I002-24848).

**Acknowledgments:** We would like to thank all students who participated in this research, as well as their teachers, school principals, and the Tosepan Titataniske Cooperative for their cooperation. This study would not have been possible without the hard work and collaboration of all the people responsible for the environmental education Community Program, including Patricia Moguel, Mayolo Hernández, Lourdes García, Octavio Zamora and José Jiménez. We thank Alma Hernández who helped with mapping. M.F.S. thanks DGAPA-UNAM for a postdoctoral fellowship; P.L.-d.-T. thanks CONACyT for the graduate fellowship.

**Conflicts of Interest:** The authors declare no conflict of interest. The funders had no role in the design of the study; in the collection, analyses, or interpretation of data; in the writing of the manuscript, or in the decision to publish the results.

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
