# Peer review of "Teenagers’ Awareness about Local Vertebrates and Their Functions: Strengthening Community Environmental Education in a Mexican Shade-Coffee Region to Foster Animal Conservation"

_sustainability, doi:10.3390/su12208684_

Round 1

Reviewer 1 Report

In the opening of the paper it would be nice to hear more about the theoretical framework used to think about community development and sustainability education.  More information about SDGs 11 and 15 should be provided for readers who are not familiar with these.

Generally, this is a well presented quantitative study, and though it is sound in its method and analysis, the design does have some limitations as a comparison study - treatment, control groups, parametric analyses, and so on.

My comment here is that the research program here would be strengthened greatly with some qualitative studies as well, where students and community members are interviewed in order to determine more contextual features of their learning and development that are hidden by the type of design used here.  If there are any aspects of the research program that are more qualitative, it would be nice to hear at some point that they do exist and will be used to enrich and extend the findings presented here.

Reviewer 2 Report

Interesting paper and insights. I am left with two questions in the background to the paper that could add value if answered but are not themselves essential.  Firstly, why the long period between the conduct of the work and the submission of a paper?  Secondly, what was the reason for the disbanding of the program after 6 years? (Longevity and adoption of a program is a significant measure of a successful assistance program).
